# Pronounced gut microbiota signatures in patients with *JAK2V617F*-positive essential thrombocythemia

Christina Schjellerup Eickhardt-Dalbøge,[1,2,3] Anna Cäcilia Ingham,[3] Henrik V. Nielsen,[3] Kurt Fuursted,[3] Christen Rune Stensvold,[3] Lee O'Brien Andersen,[3] Morten Kranker Larsen,[2,4] Lasse Kjær,[2] Sarah Friis Christensen,[2] Trine Alma Knudsen,[2] Vibe Skov,[2] Christina Ellervik,[4,5,6] Lars Rønn Olsen,[7] Hans Carl Hasselbalch,[2,4] Jens Jørgen Elmer Christensen,[1,8] Xiaohui Chen Nielsen[1]

**ABSTRACT** Essential thrombocythemia (ET) is part of the Philadelphia chromosome-negative myeloproliferative neoplasms. It is characterized by an increased risk of thromboembolic events and also to a certain degree hypermetabolic symptoms. The gut microbiota is an important initiator of hematopoiesis and regulation of the immune system, but in patients with ET, where inflammation is a hallmark of the disease, it is vastly unexplored. In this study, we compared the gut microbiota via amplicon-based 16S rRNA gene sequencing of the V3–V4 region in 54 patients with ET according to mutation status Janus-kinase 2 (*JAK2V617F*)-positive vs *JAK2V617F*-negative patients with ET, and in 42 healthy controls (HCs). Gut microbiota richness was higher in patients with ET (median-observed richness, 283.5; range, 75–535) compared with HCs (median-observed richness, 191.5; range, 111–300; $P < 0.001$). Patients with ET had a different overall bacterial composition (beta diversity) than HCs (analysis of similarities [ANOSIM]; $R = 0.063$, $P = 0.004$). Patients with ET had a significantly lower relative abundance of taxa within the *Firmicutes* phylum compared with HCs (51% vs 59%, $P = 0.03$), and within that phylum, patients with ET also had a lower relative abundance of the genus *Faecalibacterium* (8% vs 15%, $P < 0.001$), an important immunoregulative bacterium. The microbiota signatures were more pronounced in patients harboring the *JAK2V617F* mutation, and highly similar to patients with polycythemia vera as previously described. These findings suggest that patients with ET may have an altered immune regulation; however, whether this dysregulation is induced in part by, or is itself inducing, an altered gut microbiota remains to be investigated.

**IMPORTANCE** Essential thrombocythemia (ET) is a cancer characterized by thrombocyte overproduction. Inflammation has been shown to be vital in both the initiation and progression of other myeloproliferative neoplasms, and it is well known that the gut microbiota is important in the regulation of our immune system. However, the gut microbiota of patients with ET remains uninvestigated. In this study, we characterized the gut microbiota of patients with ET compared with healthy controls and thereby provide new insights into the field. We show that the gut microbiota of patients with ET differs significantly from that of healthy controls and the patients with ET have a lower relative abundance of important immunoregulative bacteria. Furthermore, we demonstrate that patients with *JAK2V617F*-positive ET have pronounced gut microbiota signatures compared with *JAK2V617F*-negative patients. Thereby confirming the importance of the underlying mutation, the immune response as well as the composition of the microbiota.

**KEYWORDS** gut microbiota, gut microbiome, essential thrombocythemia, myeloproliferative neoplasms, ET, polycythemia vera, myelofibrosis, *JAK2V617F*, inflammation

Address correspondence to Christina Schjellerup Eickhardt-Dalbøge, christinadalboege@gmail.com.

Hans Carl Hasselbalch, Jens Jørgen Elmer Christensen, and Xiaohui Chen Nielsen contributed equally to this article.

H.C.H. has received research funding from Novartis and is on the data monitoring board for AOP Orphan. All other authors declare no conflict of interest.

See the funding table on p. 15.

Essential thrombocythemia (ET) belongs to the Philadelphia chromosome-negative myeloproliferative neoplasms (MPNs), which also include polycythemia vera (PV), primary myelofibrosis (PMF), and prefibrotic PMF (1, 2). ET is characterized by thrombocytosis of $>450 \times 10^9$/L in the peripheral blood, and symptoms include thrombosis, erythromelalgia, fatigue, and bleedings (3, 4). In patients with ET, 50%–60% has a mutation in the gene encoding the Janus-kinase 2 (*JAK2V617F*) (5), 20%–25% has a mutation in the endoplasmic reticulum-associated protein calreticulin (*CALR*) gene (6, 7), and 5% has a mutation in the myeloproliferative leukemia virus oncogene (*MPL*) encoding the thrombopoietin receptor (8). These mutations are associated with different risk profiles, with patients with *JAK2V617F* mutation having the highest risk of thrombosis compared to patients with the *CALR* mutation (4, 6, 9). Both the *JAK2V617F* mutation and the *CALR* mutation induce a chronic inflammatory state by the generation of reactive oxygen species (10–13) and increased interleukin-6 (IL-6) production (14, 15), respectively.

The gut microbiota (the bacteria, fungi, archaea, viruses, and protozoa present in the gut) is important for the metabolism of food and regulation of the immune system. Changes in the gut microbiota are associated with several illnesses such as autoimmune diseases (16), depression (17), non-alcoholic fatty liver disease, obesity, and type 2 diabetes (18). Furthermore, the gut microbiota modulates side effects and response to chemotherapy (19–21), and in patients with acute myeloid leukemia (AML), a low abundance of *Faecalibacterium* was associated with poor prognosis possibly due to a damaged intestinal barrier (22). In addition, it is well established that the gut microbiota is an important initiator of hematopoiesis (23), and a gut-bone marrow axis has been proposed (24).

We have previously shown that patients with PV have a higher observed bacterial richness, higher abundance of *Bacteroides*, and a lower abundance of taxa within the *Firmicutes* phylum, including *Faecalibacterium*, compared with healthy controls (HCs) (25). Oliver et al. (26) showed that patients with MPN (*N* = 25), including eight patients with ET, had a lower relative abundance of *Parabacteroides* compared with HCs, but the microbiota of patients with ET was not investigated separately. Yoon et al. showed that the thrombocyte count and the relative abundance of *Faecalibacterium* were inversely associated in subjects in the general population (27).

No previous studies have investigated the gut microbiota in patients with ET alone. In this study, we investigated the gut microbiota of 54 patients with ET compared with 42 HCs using amplicon sequencing of the 16S rRNA gene. Furthermore, we investigated whether the microbiota differed according to mutation status (*JAK2V617F* positivity vs *JAK2V617F* negativity).

## RESULTS

### Patient and healthy control characteristics

In total, 54 patients with ET and 42 HCs fulfilled the inclusion criteria. No significant difference was observed in age, with a median age of 68 years (range, 38–83) in the patients with ET vs 71 years (range, 66–74) in the HCs (Table 1). Patients with ET were more often female (66.7% vs 40.5%, *P* = 0.013) and had a higher Charlson comorbidity index (CCI) (median, 1; range, 0–4 vs median, 0; range, 0–4; *P* = 0.038) compared with the HCs. More patients with ET had hypertension (61.1% vs 38.1%, *P* = 0.039). Thirty-six patients harbored the *JAK2V617F* mutation, seven the *CALR* mutation, three the *MPL* mutation, six were triple negative, and two patients were positive for both *CALR* and *JAK2V617F* (Fig. 1A). No significant differences were observed in terms of smoking, body mass index (BMI), or leukocyte count between patients with ET and HCs; however, a significant difference was observed in thrombocyte count, hematocrit, and estimated glomerular filtration rate (eGFR) (Table 1).

**TABLE 1** Baseline characteristics of patients with ET vs HCs[d]

| Characteristics | Group | | P value |
| --- | --- | --- | --- |
| | ET | HCs | |
| Number of patients | 54 | 42 | |
| Sex, N (%) | | | |
| Female | 36 (66.7) | 17 (40.5) | 0.013 |
| Age,[b] median (range) | 68 (38–83) | 71 (66–74) | 0.12 |
| Mutation, N (%) | | | |
| CALR | 7 (13) | 0 | |
| JAK2V617F | 36 (66.7) | 0 | |
| MPL | 3 (5.6) | NA | |
| Triple negative | 6 (11.1) | 0 | |
| JAK2V617F + CALR | 2 (3.7) | 0 | |
| BMI (kg/m$^2$)[c] | | | |
| mean (SD) | 25.5 (4.8) | 24.9 (2.4) | 0.5 |
| Hypertension, N (%) | | | |
| Yes | 33 (61.1) | 16 (38.1) | 0.039 |
| No | 21 (38.9) | 23 (54.7) | |
| Unknown | 0 | 3 (7.14) | |
| Charlson comorbidity index[a] | | | |
| Median (range)[b] | 1 (0–4) | 0 (0–4) | 0.038 |
| No comorbidities (CCI = 0), N (%) | 15 (27.8) | 22 (52.4) | 0.02 |
| Low burden (CCI ≤ 2), N (%) | 26 (48.1) | 16 (38.1) | 0.41 |
| Moderate to high (CCI > 2), N (%) | 13 (24.1) | 4 (9.5) | 0.1 |
| Blood test | | | |
| Leukocyte × 10$^9$/L, mean (SD)[c] | 6.2 (1.7) | 6.4 (1.1) | 0.5 |
| Hematocrit (%), mean (SD)[c] | 40 (3) | 43 (3) | <0.001 |
| Thrombocyte count × 10$^9$/L, median (range)[b] | 376 (161–903) | 227 (125–355) | <0.001 |
| eGFR, mean (SD)[c] | 81.9 (15.3) | 73.1 (13.2) | 0.014 |
| Smoking, N (%) | | | |
| Current smoker | 5 (9.3) | 1 (2.4) | 0.23 |
| Former smoker | 26 (48.1) | 13 (31) | 0.1 |
| Never smoker | 22 (40.1) | 26 (61.9) | 0.06 |
| Unknown | 1 (1.9) | 2 (4.8) | |
| Treatment, N (%) | | | |
| IFN | 11 (20.4) | Not relevant | |
| HU | 23 (42.6) | | |
| No treatment | 15 (27.8) | | |
| Other | 5 (9.3) | | |

[a]Comorbidity scores were calculated using CCI.
[b]Pairwise Wilcoxon rank sum test.
[c]Analysis of variance (ANOVA) with *post hoc* Tukey honest significant difference (HSD) test.
[d]ns, not significant; IFN, interferon-alpha; HU, hydroxyurea; NA, not applicable.

## Higher observed richness in patients with ET than in HCs

Patients with ET had significantly higher bacterial richness (median-observed richness, 283.5; range, 75–535) compared with HCs (median-observed richness, 191.5; range, 111–300; $P < 0.001$) (Fig. 1B); however, no significant difference was observed in alpha diversity between the patients with ET and HCs (median Inverse Simpson 22.9 vs 26.3) (Fig. 1C).

## Composition of the gut microbiota and relative abundance in patients with ET are different from those in HCs

The overall gut bacterial composition (beta diversity) differed significantly between patients with ET and HCs (ANOSIM; $R = 0.063$, $P = 0.004$) (Fig. 2A). To identify the

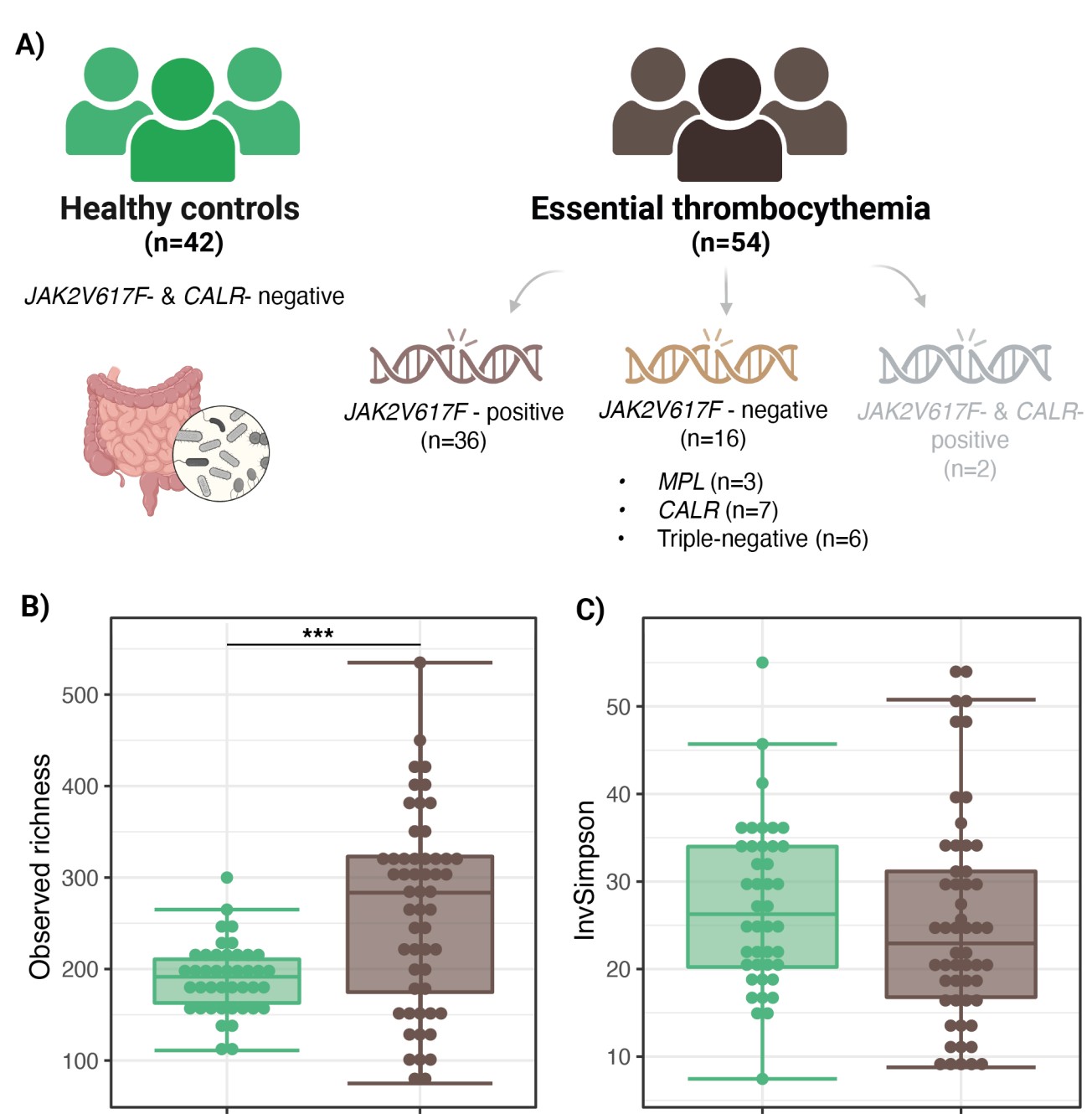

**FIG 1** Study design, observed richness, and alpha diversity (Inverse Simpson index) of patients with ET and HCs. (A) Overview of the study design and mutation status in the 42 HCs and 54 patients with ET. (B) The observed richness [amplicon sequence variants (ASVs)] in patients with ET vs HCs is illustrated. Patients with ET had a higher observed richness (median-observed richness, 283.5; range, 75–535) compared with HCs (median-observed richness, 191.5; range, 111300; P < 0.001). (C) The alpha diversity (Inverse Simpson index) of the patients with ET compared with HCs. No significant difference was found in alpha diversity between the HCs and patients with ET.

genera, explaining these compositional differences, we employed a principal component analysis (PCA), identifying *Bacteroides*, *Blautia*, *Prevotella_9*, *Akkermansia*, and *Faecalibacterium* to contribute most to the variation between groups, as illustrated in the biplot in Fig. 2A.

Several gut microbial taxa differed between patients with ET compared with HCs as identified using LEfSe (Fig. 2B). Patients with ET had a significantly lower relative

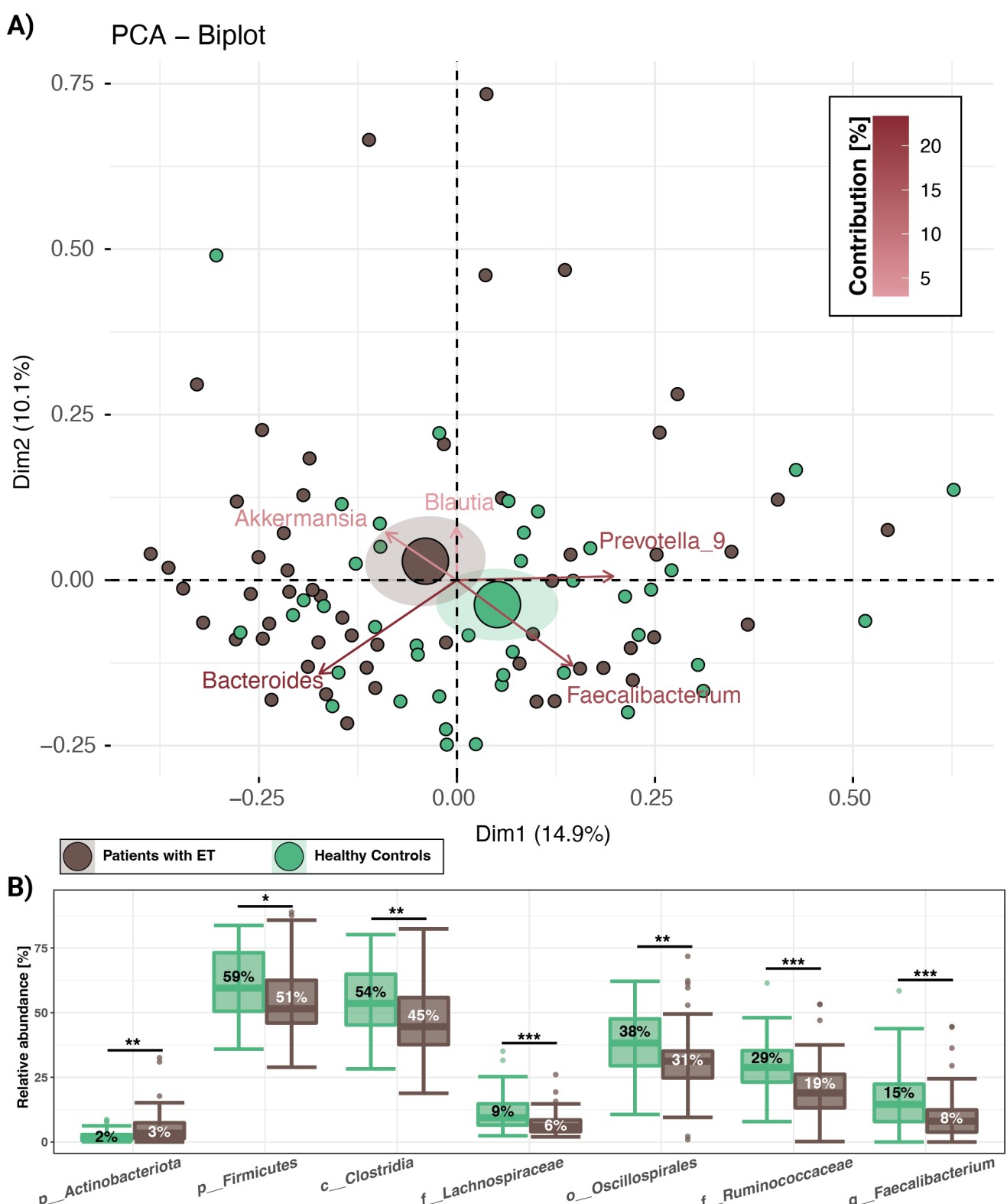

**FIG 2** PCA and differential abundance analysis of the gut microbiota in patients with ET compared with HCs. (A) PCA of the patients with ET compared with HCs. The top five predictors are shown in the PCA biplot at genus level, and the red arrows mark their contribution, i.e., the percentage of variance between the groups explained by each genus. (B) Differential abundance analysis of the gut microbiota in patients with ET compared with HCs. Linear discriminant analysis Effect Size (LEfSe) was used to identify significantly different taxa between the groups. Only taxa with an overall median proportion >1% and a linear discriminant analysis (LDA) score >0.005 are shown. Abbreviations: p, phylum; c, class; o, order; f, family; g, genus. *P < 0.05, **P < 0.01, and ***P < 0.001.

abundance of taxa within the *Firmicutes* phylum compared with HCs (51% vs 59%, *P* = 0.03). Patients with ET had a significantly lower relative abundance of class *Clostridia* (45% vs 54%, *P* = 0.001), e.g., the *Lachnospiraceae* family (6% vs 9%, *P* < 0.001), compared with HCs. Furthermore, patients with ET had a significantly lower relative abundance of the order *Oscillospirales* (31% vs 38%, *P* < 0.01) compared with HCs, especially the *Ruminococcaceae* family (19% vs 29%, *P* < 0.001), and the genus *Faecalibacterium* (8% vs 15%, *P* < 0.001). The majority (89%) of reads affiliated with the genus *Faecalibacterium* had a species annotation as *F. prausnitzii*, the rest were unspecified species.

These findings were confirmed by MaAsLin two analysis for all significantly different taxa except for *Firmicutes*, where MaAsLin two analysis, after correcting for multiple testing, only showed a tendency, *P* = 0.067.

## Clustering according to bacterial composition

To further investigate the microbiota differences, we assessed taxonomical clustering. All samples were grouped into three clusters (Fig. 3). Cluster 1 was dominated by the genera *Bacteroides* (mean relative abundance 21%) and *Faecalibacterium* (mean relative abundance 16%). Cluster 2 was dominated by the genera *Bacteroides* (mean relative abundance 31%) and *Subdoligranulum* (mean relative abundance 11%). Cluster 3 consisted of the remaining samples that did not fit into cluster 1 or cluster 2, and several different genera were equally represented (*Blautia*, *Faecalibacterium*, *Bacteroides*, and *Veillonella*) (Fig. 3). The majority of HCs (79%, *N* = 33) belonged to cluster 1, 14% (*N* = 6) grouped into cluster 2, and 7% (*N* = 3) into cluster 3. Patients with ET were distributed among the three clusters [57% in cluster 1 (*N* = 31), 35% in cluster 2 (*N* = 19), and 7% in cluster 3 (*N* = 4)].

Hence, a similar number of patients with ET and HCs grouped into cluster 1 (48% [*N* = 31] vs 52% [*N* = 33]) and cluster 3 (57% [*N* = 4] vs 43% [*N* = 3]), while cluster 2 was populated by more samples from patients with ET (76% [*N* = 19] vs 24% [*N* = 6] HC samples).

This confirmed the high abundance of *Faecalibacterium* in HCs but also illustrated compositional differences within patients with ET, and that not all patients with ET differ equally from HCs in their bacterial composition.

## Demographic data and characteristics according to mutation

To investigate if the gut microbiota differed according to mutation status of patients with ET, the patients were divided into *JAK2V617F*-positive patients vs *JAK2V617F*-negative patients (*CALR*, *MPL*, or triple negative). Thirty-eight patients were *JAK2V617F* positive, nine were *CALR* mutated, three were positive for the *MPL* mutation, and six were

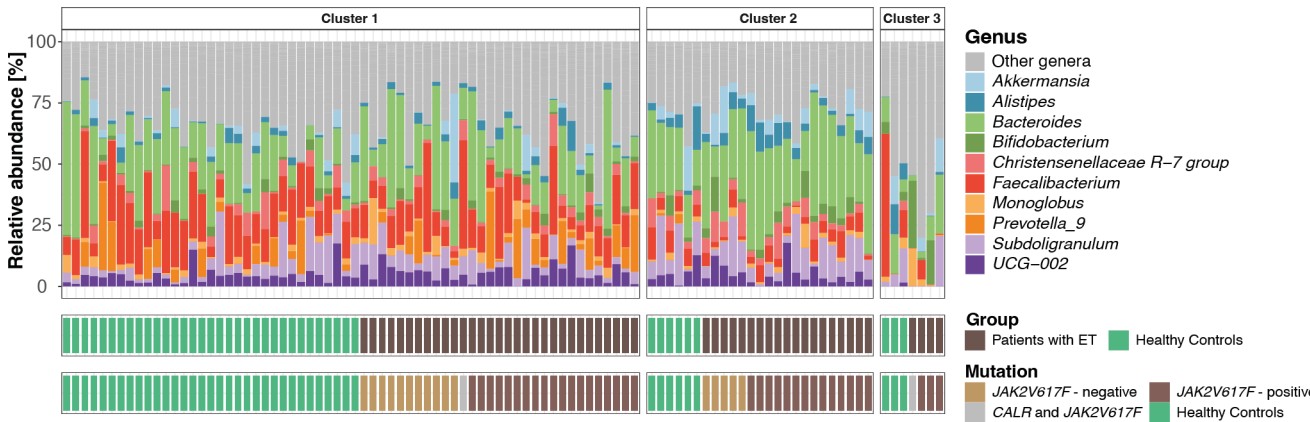

**FIG 3** Clustering according to bacterial composition of the gut microbiota in patients with ET and HCs. In the first panel, the relative abundances per sample of the top 10 genera of The gut microbiota in the three clusters are shown. The panels in the second and third row show the group affiliation (patient with ET, HCs, *JAK2V617F*-positive, *JAK2V617F*-negative patients, and patients positive for both *JAK2V617F* and *CALR*).

triple negative. Two patients were positive for both the *JAK2V617F* and *CALR* and were excluded from these analyses; thus, 36 patients were left in the *JAK2V617F* group and 16 in the group without the *JAK2V617F* mutation (Fig. 1A). The demographic data of the two groups are illustrated in Table S1. Patients positive for the *JAK2V617F* mutation had a higher frequency of women (69.4% vs 40.5%, *P* = 0.039), were more likely to have hypertension (69.4% vs 38.1%, *P* = 0.021), and had a higher comorbidity burden (median CCI 1, range 0–4 vs median CCI 0, range 0–4, *P* = 0.02) compared with the HCs. No significant differences were seen according to sex, hypertension, and comorbidities when the *JAK2V617F*-positive patients were compared with the *JAK2V617F*-negative patients. Patients without the *JAK2V617F* mutation had a tendency to be younger compared with the HCs (median age 64.5 years, range 46–81). For further details, see Table S1.

## Higher observed richness in *JAK2V617*-positive patients with ET than in HCs

Patients positive for the *JAK2V617F* mutation had a significantly higher observed bacterial richness (median-observed ASVs, 299.5; range, 102–535) compared with the HCs (median-observed ASVs, 191.5; range, 111–300; *P* < 0.001) (Fig. 4A). No significant differences in observed richness were found when *JAK2V617F*-negative patients (median-observed ASVs, 233; range, 75–406) were compared with the HCs or with the *JAK2V617F*-positive patients (Fig. 4A). No significant differences in alpha diversity (inverse Simpson index) were found when the two patient groups were compared with each other or the HCs (Fig. 4B).

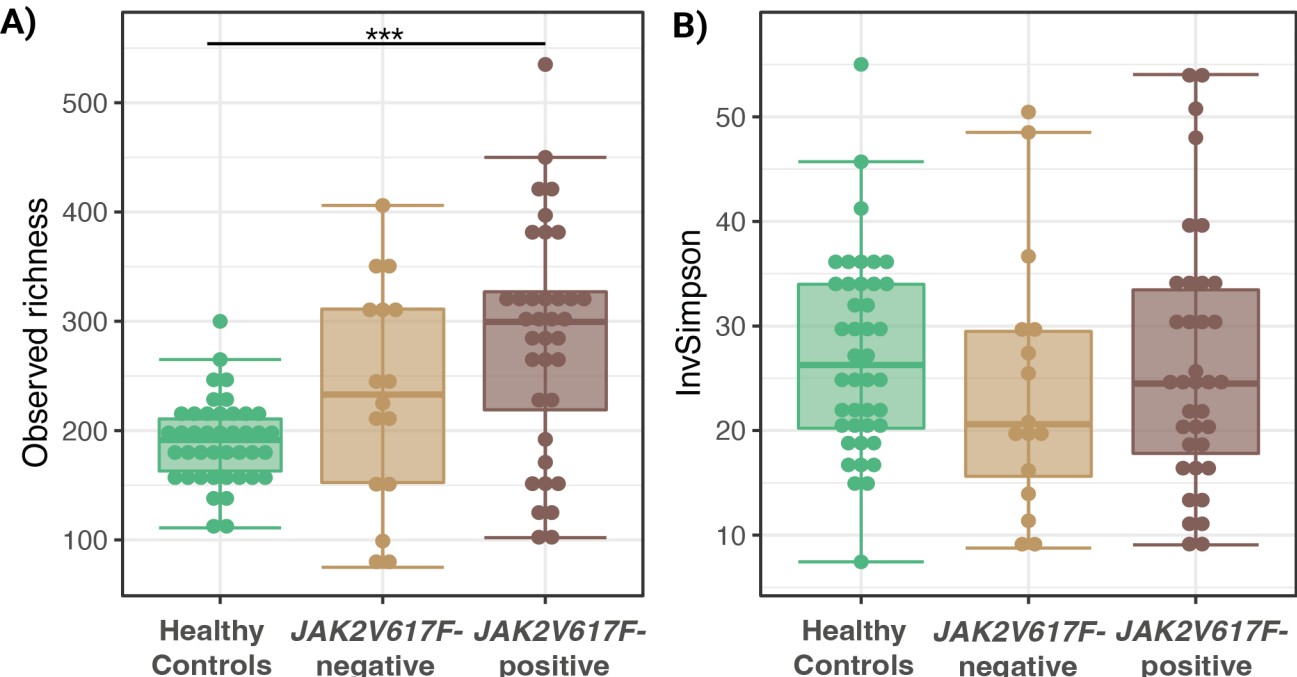

**FIG 4** Gut microbiota-observed richness and alpha diversity (inverse Simpson index) of patients with ET according to mutation status and the HCs. (A) Observed richness (ASVs) in patients with ET vs HCs, *JAK2V617F*-positive patients had a higher observed richness (median-observed ASVs, 299.5; range, 102–535) compared with the HCs (median-observed richness, 191.5; range, 111–300; *P* < 0.001). (B) Alpha diversity measured by inverse Simpson index in the three groups. No significant difference was found between the HCs and the two mutation groups when comparing alpha diversity measured by median inverse Simpson index. ***P* < 0.001.

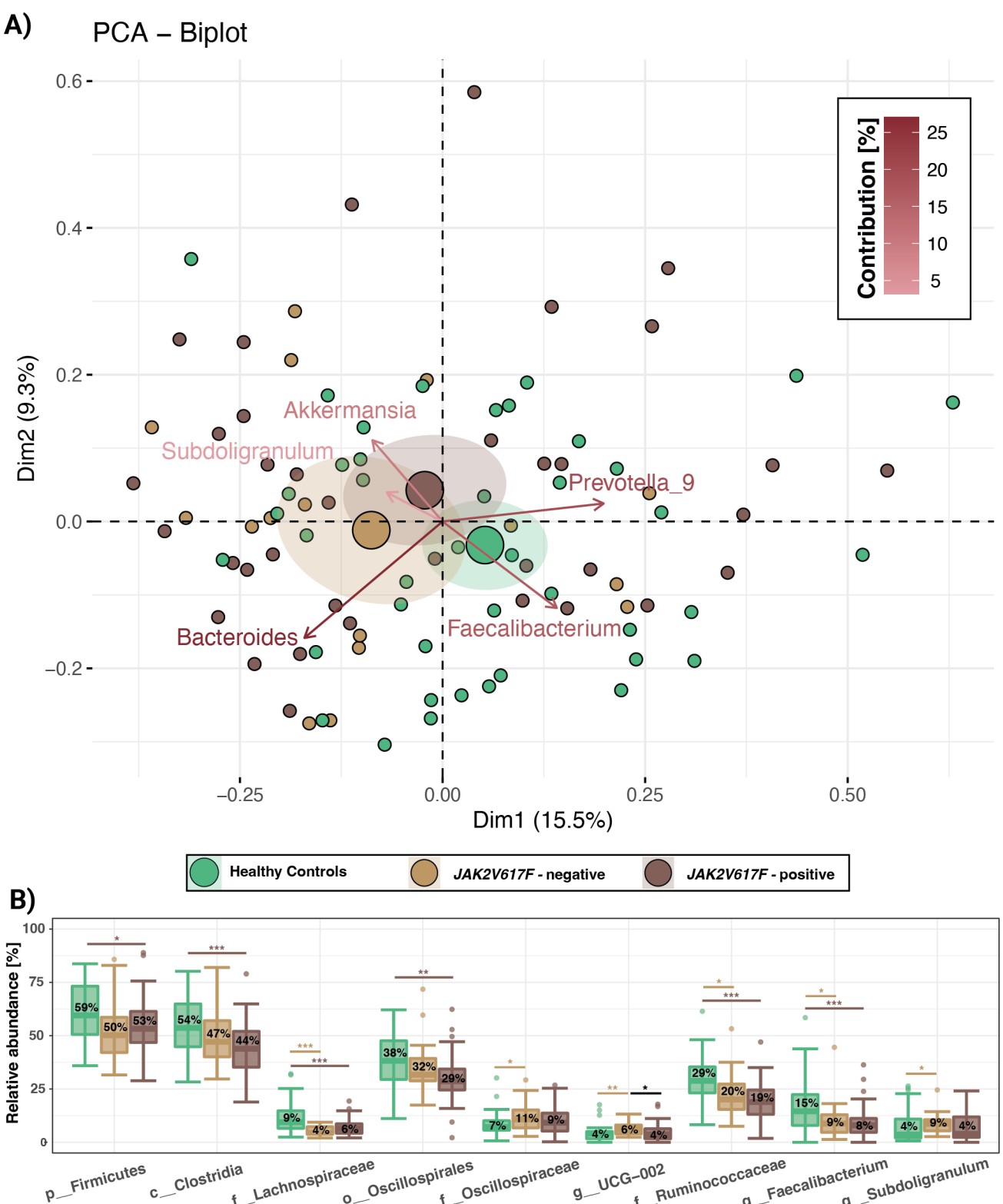

**FIG 5** PCA and differential abundance analysis of the patients with ET according to mutation status compared with HCs. (A) PCA of the patients with ET according to mutation status compared with HCs. The top five predictors are shown in the PCA biplot at genus level, and the red arrows mark their contribution. (B) LEfSe was used to identify significantly different taxa between the groups (*JAK2V617F*-positive and *JAK2V617F*-negative patients compared with HCs). Selected taxa with an overall median proportion >1% and an LDA score >0.005 are shown. Significantly different taxa are indicated with asterisks. Abbreviations: p, phylum; c, class; o, order; f, family; g, genus. *$P < 0.05$, **$P < 0.01$, and ***$P < 0.001$.

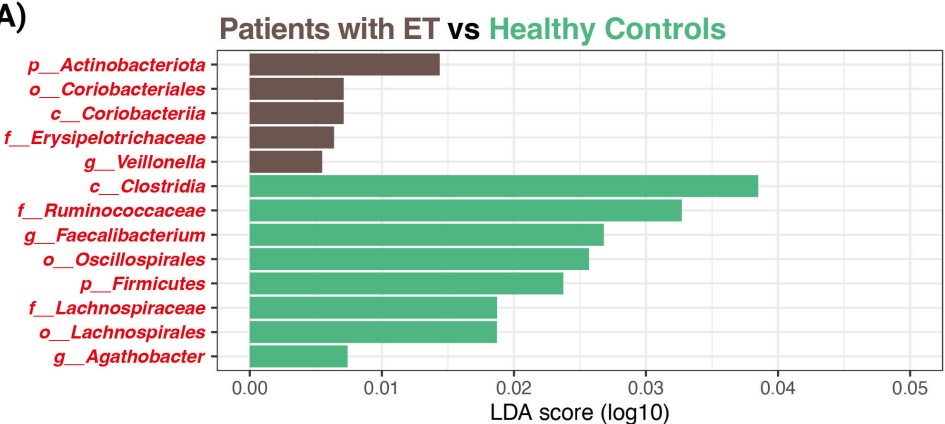

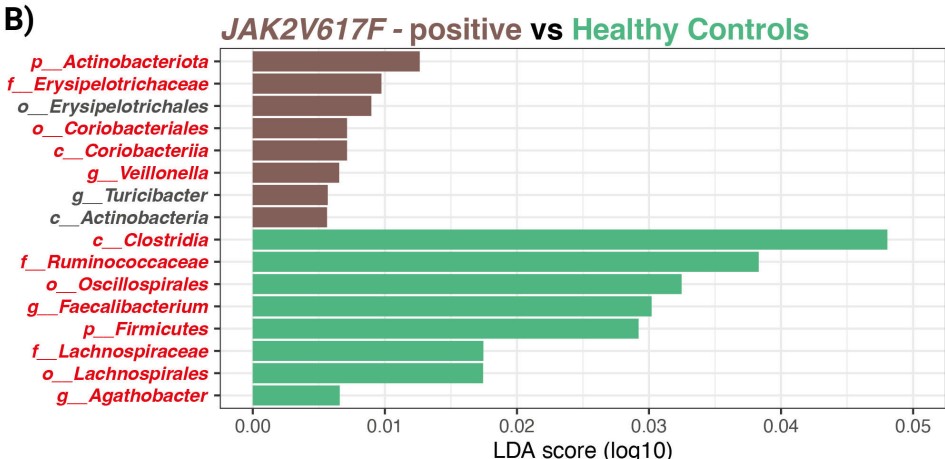

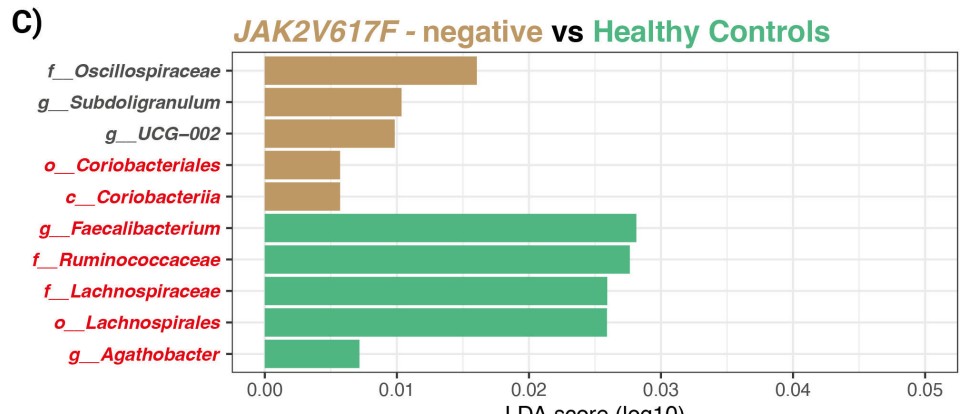

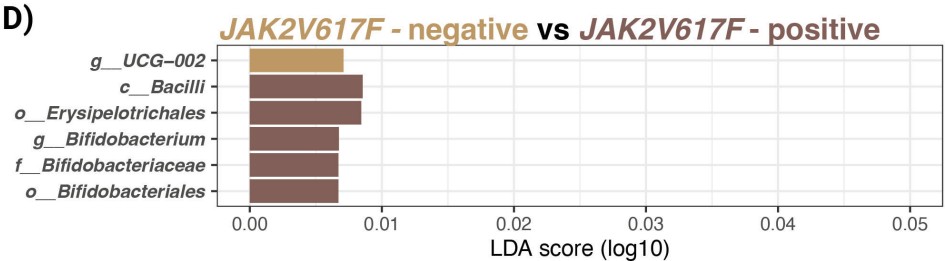

FIG 6 Differential abundance analysis of the different groups. LEfSe was used to identify significantly different taxa between the groups (patients with ET, *JAK2V617F*-positive and *JAK2V617F*-negative patients compared with HCs and each other). Only significantly different taxa with an LDA score >0.005 are shown. Bars indicate in which group each respective taxon is

**FIG 6** (Continued)

overrepresented. Bar length corresponds to LDA score, an indicator of effect size. Taxa written in red differed significantly when the whole group of ET patients was compared with HCs. (A) The comparison between all patients with ET and HCs is shown. (B) The comparison between *JAK2V617F*-positive patients and HCs is shown. All the taxa that differed between the patients with ET and HCs (red), and a few extra differed when *JAK2V617F*-positive patients were compared with HCs. When the *JAK2V617F*-negative patients were compared with the HCs (C) fewer taxa differed and with a lower LDA score. (D) The *JAK2V617F*-positive patients are compared with the *JAK2V617F*-negative patients. Abbreviations: p, phylum; c, class; o, order; f, family; g, genus.

## Composition of the gut microbiota in patients with ET according to mutation group compared with HCs

Patients who harbored the *JAK2V617F* mutation differed significantly in overall gut microbiota composition compared with the HCs (ANOSIM; $P = 0.001$, $R = 0.096$). The top five contributing genera are shown in Fig. 5A. No significant difference was found between *JAK2V617F*-negative patients and the HCs (ANOSIM; $P = 0.08$, $R = 0.109$), or between the two patient groups.

Differential abundance analysis by LEfSe showed that several taxa within the phylum *Firmicutes* were less abundant in *JAK2V617F*-positive patients compared with the HCs. Overall, the same taxa differed between the *JAK2V617F*-positive patients compared with the HCs as when the whole group of patients with ET was compared with HCs (Fig. 5B and Fig. 6; Fig. S1).

When the *JAK2V617F*-negative patients were compared with the HCs, some of the same taxa differed as when the *JAK2V617F*-positive patients were compared with the HCs. The *JAK2V617F*-negative patients, like the *JAK2V617F*-positive patients, had a lower relative abundance of *Lachnospiraceae* (4% vs 9%, $P < 0.001$) and *Ruminococcaceae* (20% vs 29%, $P = 0.025$) at family level, and *Faecalibacterium* (9% vs 15%, $P = 0.016$) at genus level. However, no significant difference was found in *Firmicutes* at phylum level, *Clostridia* at class level, and *Oscillospirales* at order level, and the differences found were less pronounced (Fig. 5B and Fig. 6; Fig. S1).

In contrast to *JAK2V617F*-positive patients, *JAK2V617F*-negative patients had a higher relative abundance of the family *Oscillospiraceae* (11% vs 7%, $P = 0.04$) and the genus *Subdoligranulum* belonging to the family *Ruminococcaceae* (9% vs 4%, $P = 0.032$) compared with the HCs (Fig. 5B and 6; Fig. S1)

When the two patient groups were compared with each other, a small but significant difference was found in the relative abundance of *UCG-002*, a genus belonging to *Oscillospiraceae* (4% in the *JAK2V617F*-positive patients vs 6% in the *JAK2V617F*-negative patients, $P = 0.035$). Patients with the *JAK2V617F* mutation had a higher relative abundance of *Bifidobacterium* compared with patients without, the difference however, was small (Fig. 6; Fig. S1).

## DISCUSSION

In this study, we compared 54 patients with ET with 42 HCs and found that patients with ET have a distinct microbiota signature compared with HCs. Although patients with ET did not differ in alpha diversity (Inverse Simpsons) *per se*, patients with ET had a significantly higher observed bacterial richness compared with the HCs. This possibly indicates a more transient microbiota with additional taxa joining the disrupted residential microbiota. Furthermore, patients with ET differed significantly in bacterial composition and several taxa were less abundant in patients with ET compared with HCs. In addition, the microbiota signatures of the *JAK2V617F*-positive patients were more pronounced than the *JAK2V617F*-negative patients.

We have previously described how the gut microbiota in patients with PV differs from that of HCs and varies according to treatment (25). To our knowledge, this is the first study to investigate the gut microbiota in patients with ET alone. Interestingly, several of the taxa, which differed when patients with ET were compared with HCs, were the same

taxa that differed in patients with PV (*Firmicutes*, *Clostridia*, *Lachnospirales*, *Oscillospirales*, *Ruminococcaceae*, and *Faecalibacterium*). This suggests that the gut microbiota in patients with MPN already differs in the earlier stages of the disease.

*JAK2V617F*-negative patients with ET did not differ according to observed richness and beta diversity compared with the HCs; however, the group of *JAK2V617*-negative patients with ET was small and the variation within the group was large. Furthermore, the group of patients with *JAK2V617F*-negative ET consisted of *CALR*, *MPL*, and triple-negative patients, and a much larger, more homogenous group would be required to draw any firm conclusions. Although the microbiota of patients without the *JAK2V617F* mutation appeared overall more similar to the HCs, some taxa still differed in abundance between the two groups. Since the microbiota differences are very similar in *JAK2V617F*-positive patients with ET and in patients with PV [of which 96% harbor the *JAK2V617F* mutation (5)], it could be speculated that the *JAK2V617F* mutation drives these specific differences in the gut microbiota. As noted above, *F. prausnitzii* had a lower relative abundance in patients with ET, which is very similar to our previously reported findings in patients with PV (25).

Taking into account the role of chronic inflammation in disease progression in the biological MPN continuum, it is highly intriguing that a low abundance of *F. prausnitzii* has been found to be associated with several chronic inflammatory diseases (16, 28). *F. prausnitzii* is a strictly anaerobic, non-spore forming, rod known to consume acetate and produce anti-inflammatory molecules (such as shikimic and salicylic acids) (29–31). Furthermore, *F. prausnitzii* is one of the main butyrate producers, and, as mentioned above, a low relative abundance is observed in patients with chronic autoimmune and inflammatory diseases such as type 2 diabetes, inflammatory bowel disease, multiple sclerosis, non-alcoholic fatty liver disease, neurodegenerative disorders, and many others (28). The metabolites of *Faecalibacterium* are important for the regulation of the intestinal barrier and are known to reduce the gut permeability (32). Therefore, a decreased abundance of *F. prausnitzii* in both ET and, as previously shown in PV (25), could possibly indicate higher gut permeability. In a study investigating extracellular vesicles and gut microbiota (33), it was found that patients with PV had an increased proportion of lipopolysaccharide (LPS)-associated extracellular vesicles, suggesting increased intestinal permeability; however, no significant differences in the gut microbiota in patients with PV were found compared with HCs (33). In another study, increased gut permeability was associated with higher LPS concentrations in the blood; furthermore, an increased LPS concentration was shown to exacerbate leukemia burden and to shorten overall survival in mice with AML (22). This further exemplifies the importance of the gut microbiota in hematological malignancies.

MPNs are known as chronic inflammatory disorders (10, 34–37) and it is therefore not unlikely that the microbiota participates in the vicious inflammatory circle leading to MPN initiation and progression. an example of this is seen in the development of pre-leukaemic myeloproliferation (PEP) in *tet methylcytosine dioxygenase 2*$^{-/-}$ mice, where microbial-dependent inflammation contributed to the development of PEP (38). So far, all studies on MPN and gut microbiota have been case-control studies, including this one; therefore, causality and directionality cannot be determined. Neither can we determine whether the gut microbiota is involved in MPN initiation, MPN progression, or both. Furthermore, studies are needed to elucidate these mechanisms.

This study had several limitations. First, due to small sample sizes, we were unable to reconcile our microbiota findings with the different treatment modalities in ET. We have previously shown that patients with PV treated with interferon-alpha have a gut microbiota more similar to HCs compared with patients with PV not treated with interferon-alpha (25), and treatment with hydroxyurea has recently been reported to be associated with an increase in the abundance of beneficial bacteria, such as *Faecalibacterium* in children with sickle cell anemia (39). Furthermore, the number of *JAK2V617F*-negative patients with ET was small (limiting the statistical power) and heterogeneous with other mutations (*CALR* and *MPL*) or triple negative. We do not know whether the

*CALR* mutation in itself has an effect on the gut microbiota, and larger studies would be needed to elucidate a possible effect. Moreover, misclassification cannot entirely be ruled out, as studies have shown that several *JAK2V617F*-positive "ET" patients are misclassified as "ET" but actually have PV when a red cell mass determination is performed using isotopic techniques (40). Of note, it is also important, but sometimes challenging, to distinguish between ET and pre-PMF. Pre-PMF often mimics ET but the prognosis differs significantly from ET, with ET having a more favorable prognosis (41–43) and possible differences in the gut microbiota. Diet and sex differences are known to influence the gut microbiota. Unfortunately, data on patients' diet were not available for this study. Furthermore, the number of patients was too small to investigate sex differences (44, 45). Large-scale studies are needed to investigate the possible role of the gut microbiota in patients with MPNs. These studies should include patients with other mutations than *JAK2V617F* (the *CALR* and *MPL* mutations), and patients with different treatment regimens. In addition, the role of the gut microbiota in the continuum from the very early disease state (clonal hematopoiesis of indeterminate potential [CHIP]) toward overt MPN development should be investigated.

In conclusion, patients with ET have a gut microbiota that differs from that of HCs in having a higher observed bacterial richness, a significantly different overall gut bacterial composition and lower relative abundances of several taxa within the phylum *Firmicutes* that may be important for maintaining a balanced inflammatory state. Furthermore, patients with *JAK2V617F*-positive ET had more pronounced gut microbiota signatures when compared with HCs, and almost the same signature as we previously found in patients with PV, possibly pointing to a *JAK2V617F*-positive MPN microbiota signature. In the future, CHIP screening for the *JAK2V617F* mutation might be supplemented by the characterization of the gut microbiota as an additional biomarker for risk assessment of MPN development, thus aiding treatment strategy decisions.

## Materials and Methods

### Patient recruitment

Patients above 18 years of age with ET according to the World Health Organization 2016 classification of MPNs (2) were included in the study, at the Department of Hematology, Zealand University Hospital, Roskilde, Denmark, from November 2018 until August 2021. Patients were excluded in case of any change of cytoreductive treatment within 3 months, use of antibiotics within 2 months, treatment with glucocorticoids, pregnancy, or inability to understand oral or written information. Patients were divided according to mutation status [*JAK2V617F* positive vs *JAK2V617F* negative (*CALR*, *MPL*, and triple negative)]. Two patients harboring both the *JAK2V617F* and *CALR* mutation were excluded when *JAK2V617F*-positive patients were compared with *JAK2V617F*-negative patients (Fig. 1A).

### Healthy control recruitment

Healthy controls (*N* = 42) were recruited during 2021 from the Danish General Suburban Population Study (GESUS), Region Zealand (25, 46) and matched 1:4 by gender and sex. None of the HCs harbored the *JAK2V617F* or *CALR* mutations at the time of entering GESUS between 2010 and 2013. HCs were excluded if they had used antibiotics within 2 months or if they showed elevated blood cell counts when the stool samples were collected in 2021.

### Sample collection

Study participants collected stool samples according to detailed written instructions. Stool samples, 2–5 mL, were collected a few hours before an outpatient visit. If the patient or HC was not able to come to the clinic, the stool sample was collected at the

house of the participant. No later than 6 hours after collection, the sample was stored at −80°C.

## Clinical and laboratory data

Biochemical and clinical data were obtained retrospectively for each patient and HC. Hemoglobin concentration (mmol/L), leukocyte and differential count ($\times10^9$/L), and thrombocyte count ($\times10^9$/L) were measured in EDTA stabilized whole blood using Sysmex XN-9000 (Sysmex Corporation). Lactate dehydrogenase (U/L), creatinine (μmol/L), and C-reactive protein were measured in plasma using Dimension Vista 1500 system (Siemens Healthcare Inc.). Kidney function was assessed by eGFR (mL/min/1.73 $m^2$) using the Chronic Kidney Disease Epidemiology Collaboration (CKD-EPI) formula (47).

Information on mutation status of the patients with ET was obtained by quantitative PCR for *JAK2V617F* (48) and fragment analysis for *CALR* (6) and MPL was detected using next generation sequencing as described previously by Grauslund et al. (49).

The HCs were screened for *JAK2V617F* or *CALR* mutations using droplet digital PCR as previously described by Cordua et al. (46).

Clinical data included information on current and previous hematological treatment, anticoagulation therapy, use of metformin and/or statins, time since ET diagnosis, BMI ($kg/m^2$), hypertension, comorbidities (as defined by CCI) (50), and smoking status.

## 16s rDNA gene extraction, sequencing, and preprocessing

The gut microbiota was analyzed using 16S rRNA gene amplicon-based sequencing of the V3–V4 regions as described in Ring et al. (51) and Krogsgaard et al. (52). Extraction of genomic DNA was done by mixing 100 mg fecal sample with 260 μL lysis buffer in tubes with 1.4‐mm zirconium beads. Bead-beating was done for 2 min at 30 Hz using a TissueLyser II (Qiagen, Germany) followed by an automated eMAG platform (bioMérieux, France).

16S amplification was done as described in Krogsgaard et al. (52), using a modified version of the 341/806 universal prokaryotic primers (pmid 15696537) targeting the V3–V4 region of the ribosomal small subunit and PCR mastermix PrimeSTAR GXL premix (Takara bio group, San José, CA, USA). Subsequent sequencing was performed on the MiSeq platform (Illumina Inc., San Diego, CA, USA) as described in Eickhardt-Dalbøge et al. (25).

## Taxonomic classification and decontamination

As previously described in Eickhardt-Dalbøge et al. (25), DADA2 (version 1.22.0) (53) was used for inference of high-resolution ASVs, quality control, chimera removal, and taxonomic classification. The truncation lengths were set to 220 bp for forward reads and 240 for reverse reads; otherwise, default settings were used. The R package DADA2 (53) was used for taxonomical classification using the Silva reference database (version 138.1) (54). For decontamination, the *decontam* R package (55) was used based on the prevalence-based method with a threshold of 0.15 identifying five ASVs, which were subsequently removed. ASVs not classified to at least order level were removed (125 ASVs) along with ASVs belonging to Archaea (20 ASVs). Furthermore, ASVs belonging to orders *Rhizobiales*, *Rhodobacterales*, and *Rhodospirillales* (18 ASVs), as well as *Cyanobacteria*, *Planctomycetota*, and *Chloroflexi* (41 ASVs), were removed.

Based on rarefaction curves, we determined that 5,000 reads per sample would be sufficient for capturing the bacterial community present. Samples exceeding this threshold were retained for analysis. The final data set consisted of 96 samples (54 patients with ET and 42 HCs) with 6,772 unique ASVs and a median read count of 47,500 (range, 6,380–160,239).

## Statistical analysis

All statistical analyses and graphical illustrations were performed in R (version 4.0.3.) (56), and the code used for the analyses can be found at 10.6084/m9.figshare.21716558. Paraclinical, clinical, and sequence data were merged into a phyloseq object using the *phyloseq* package (57). The packages *ggplot2*, *cowplot*, and *factoextra* were used for illustrations (58–60). Probability (*P*) values ≤0.05 were considered to indicate statistical significance, and the Benjamini-Hochberg correction was used to adjust for multiple testing. ANOVA was used for continuous normally distributed data, whereas pairwise Wilcoxon rank sum tests were used when data were not normally distributed. Fisher's exact test was applied to categorical data.

We calculated alpha diversity on untransformed bacterial counts. We assessed both observed richness (the number of ASVs per sample) and the inverse Simpson index (an alpha diversity index taking into account both richness and evenness). Based on Hellinger-transformed counts, PCA was conducted to illustrate differences in beta diversity (package *FactoMineR*) (61). Differences in composition between the groups (HCs, patients with ET, *JAK2V617F*-positive patients, and *JAK2V617F*-negative patients) were assessed using analysis of similarities (ANOSIM, *vegan* package) (62) based on Bray-Curtis dissimilarities. To test which taxa significantly differed in relative abundance between the groups, we employed two differential abundance analysis methods. LEfSe at class, order, family, and genus level was used with a LDA cutoff of 0.005 (*microbiome-Marker* package) (63). To confirm the results, we also employed linear models in MaAsLin 2 (microbiome multivariate associations with linear models) for taxa with ≥1% relative abundance in ≥30% of samples with Benjamini-Hochberg correction for multiple testing (64).

Clustering was done as previously described in Ingham et al. (65), by grouping samples from HCs and patients with ET into clusters according to the composition of their bacterial communities by partitioning around medoid clustering (*cluster* package) (66). The optimal number of clusters was determined by the gap statistic, silhouette width, and elbow methods from the *factoextra* package (59).

## ACKNOWLEDGMENTS

First of all, we convey our gratitude to all the participants of the study. Furthermore, we convey our gratitude to all the doctors at the Department of Hematology, Zealand University Hospital, Roskilde, Denmark, who helped include patients to the study, and the lab technicians at Regional Department of Clinical Microbiology, University Hospital of Region Zealand, Slagelse, Denmark, and the Department of Bacteria, Parasites and Fungi, Statens Serum Institut, Copenhagen, Denmark, for their help with analyzing and dividing the samples. We are also grateful for Mette Grymer Jensens (medical and research secretary) help with organizing and collection of stool samples.

This work was founded by the Department of Bacteria, Parasites and Fungi, Statens Serum Institut, Copenhagen, Denmark, The Region Zealand Foundation for Health Research, and The Danish Cancer Society. C.E. is partly funded by the Laboratory Medicine Endowment Fund of Boston Children's Hospital.

H.C.H. has received research funding from Novartis and is on the data monitoring board for AOP Orphan. All other authors declare no conflict of interest.

H.C.H., J.J.E.C., X.C.N., L.K., V.S., H.V.N., K.F., and C.S.E.-D. designed the study. H.C.H. and C.S.E.-D. included the patients with ET. J.J.E.C., H.C.H., X.C.N., T.A.K., L.K., V.S., K.F., H.V.N., and C.S.E.-D. raised the funding. C.E., M.K.L., and C.S.E.-D. included the healthy controls. A.C.I. and L.O.A. did the bioinformatics. A.C.I. and C.S.E.-D. did the statistical analysis, tables, and data visualization. The results were interpreted by A.C.I., C.S.E-D., H.C.H., J.J.E.C., X.C.N., V.S., H.V.N., C.R.S., and K.F. C.S.E-D and A.C.I. wrote the initial draft, and all authors made a substantial contribution to the revision of the draft. The final version was approved by all authors.

## AUTHOR AFFILIATIONS

[1]Regional Department of Clinical Microbiology, Zealand University Hospital, Koege, Denmark

[2]Department of Hematology, Zealand University Hospital, Roskilde, Denmark

[3]Department of Bacteria, Parasites and Fungi, Statens Serum Institut, Copenhagen, Denmark

[4]Faculty of Health and Medical Sciences, University of Copenhagen, Copenhagen, Denmark

[5]Department of Laboratory Medicine, Boston Children's Hospital, Harvard Medical School, Boston, Massachusetts, USA

[6]Department of Data and Data Support, Region Zealand, Sorø, Denmark

[7]Department of Health Technology, Technical University of Denmark, Lyngby, Denmark

[8]Institute of Clinical Medicine, University of Copenhagen, Copenhagen, Denmark

## AUTHOR ORCIDs

Christina Schjellerup Eickhardt-Dalbøge http://orcid.org/0000-0002-0740-0555
Anna Cäcilia Ingham http://orcid.org/0000-0001-6079-6643
Henrik V. Nielsen http://orcid.org/0000-0002-6773-1874
Kurt Fuursted http://orcid.org/0000-0002-3483-3145
Christen Rune Stensvold http://orcid.org/0000-0002-1417-7048
Lee O'Brien Andersen http://orcid.org/0009-0009-6266-6514
Lasse Kjær http://orcid.org/0000-0001-6767-0226
Sarah Friis Christensen http://orcid.org/0000-0003-2178-0440
Trine Alma Knudsen http://orcid.org/0000-0001-9829-3099
Vibe Skov http://orcid.org/0000-0003-0097-7826
Christina Ellervik http://orcid.org/0000-0002-3088-4375
Hans Carl Hasselbalch http://orcid.org/0000-0003-3936-8032
Jens Jørgen Elmer Christensen http://orcid.org/0000-0001-6721-4917

## FUNDING

| Funder | Grant(s) | Author(s) |
|---|---|---|
| The Region Zealand Foundation for health Research | | Jens Jørgen Elmer Christensen |
| The Department of Bacteria, Parasites and Fungi, SSI, Copenhagen | | Jens Jørgen Elmer Christensen |
| The Danish Cancer Society | | Jens Jørgen Elmer Christensen |

## AUTHOR CONTRIBUTIONS

Christina Schjellerup Eickhardt-Dalbøge, Conceptualization, Data curation, Formal analysis, Funding acquisition, Investigation, Methodology, Project administration, Visualization, Writing – original draft, Writing – review and editing | Anna Cäcilia Ingham, Data curation, Formal analysis, Investigation, Resources, Supervision, Visualization, Writing – original draft, Writing – review and editing | Henrik V. Nielsen, Conceptualization, Data curation, Funding acquisition, Investigation, Methodology, Resources, Supervision, Writing – review and editing | Kurt Fuursted, Conceptualization, Funding acquisition, Methodology, Resources, Supervision, Writing – review and editing | Christen Rune Stensvold, Resources, Supervision, Writing – review and editing | Lee O'Brien Andersen, Data curation, Resources, Writing – review and editing | Morten Kranker Larsen, Data curation, Investigation, Writing – review and editing | Lasse Kjær, Conceptualization, Funding acquisition, Methodology, Writing – review and editing | Sarah Friis Christensen, Conceptualization, Writing – review and editing | Trine Alma Knudsen, Conceptualization, Writing – review and editing | Vibe Skov, Conceptualization, Funding

acquisition, Methodology, Writing – review and editing | Christina Ellervik, Data curation, Investigation, Writing – review and editing | Lars Rønn Olsen, Writing – review and editing | Hans Carl Hasselbalch, Conceptualization, Data curation, Funding acquisition, Investigation, Project administration, Resources, Supervision, Writing – review and editing | Jens Jørgen Elmer Christensen, Conceptualization, Funding acquisition, Investigation, Methodology, Project administration, Resources, Supervision, Writing – review and editing | Xiaohui Chen Nielsen, Conceptualization, Funding acquisition, Investigation, Methodology, Project administration, Resources, Supervision, Writing – review and editing

## DATA AVAILABILITY

The sequence data for patients with ET, anonymized, are available at the European Nucleotide Archive (ENA) (PRJEB58224); data on HCs are available upon request to the corresponding author. The STORMS Checklist (67) was completed and can be found at doi.org/10.6084/m9.figshare.22060499.

## ETHICS APPROVAL

The project was conducted in accordance with the Declaration of Helsinki and was approved by the Danish Data Protection Agency (REG-050-2015 and REG-054-2018), the National Committee on Health Research Ethics (SJ-452 and SJ-698), and the Regional Ethics Committee (SJ452). All HCs were invited by letter and thoroughly informed in person. Patients were informed in person by their physician. Written consent was obtained from all study participants.

## ADDITIONAL FILES

The following material is available online.

### Supplemental Material

**Table S1 and Fig. S1 (Spectrum00662-23-s0001.pdf).** Table S1: baseline characteristics of patients with ET versus HC. Figure S1: differential abundance analysis of the gut microbiota in patients with ET according to mutations status compared with healthy controls.

### Open Peer Review

**PEER REVIEW HISTORY (review-history.pdf).** An accounting of the reviewer comments and feedback.

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
