## [Reviewer comments · Microbiology Spectrum]

Microbiology Spectrum

Pronounced Gut Microbiota Signatures in Patients with *JAK2V617F*-positive Essential Thrombocythemia

Christina Eickhardt-Dalbøge, Anna Ingham, Henrik Nielsen, Kurt Fuursted, Christen Stensvold, Lee Andersen, Morten Larsen, Lasse Kjær, Sarah Christensen, Trine Knudsen, Vibe Skov, Christina Ellervik, Lars Olsen, Hans Hasselbalch, Jens Jørgen Christensen, and Xiaohui Nielsen

Corresponding Author(s): Christina Eickhardt-Dalbøge, The Regional Department of Clinical Microbiology, University Hospital of Region Zealand

Review Timeline:

Submission Date:	February 13, 2023
Editorial Decision:	June 26, 2023
Revision Received:	July 5, 2023
Editorial Decision:	July 17, 2023
Revision Received:	July 17, 2023
Accepted:	July 18, 2023

Editor: Jennifer Auchtung

Reviewer(s): Disclosure of reviewer identity is with reference to reviewer comments included in decision letter(s). The following individuals involved in review of your submission have agreed to reveal their identity: Modher Nagem Abed Abed (Reviewer #1); Xiangyang Kong (Reviewer #4)

Transaction Report:

DOI: <https://doi.org/10.1128/spectrum.00662-23>

June 26, 2023

Dr. Christina Schjellerup Eickhardt-Dalbøge
The Regional Department of Clinical Microbiology, University Hospital of Region Zealand
The Regional Department of Clinical Microbiology, University Hospital of Region Zealand
Ingemannsvej 46, stuen
Slagelse 4200
Denmark

Re: Spectrum00662-23 (Pronounced Gut Microbiota Signatures in Patients with *JAK2V617F*-positive Essential Thrombocythemia)

Dear Dr. Christina Schjellerup Eickhardt-Dalbøge:

Link Not Available

Sincerely,

Jennifer Auchtung

Journals Department
Reviewer comments:

Reviewer #2 (Comments for the Author):

In the present paper, the gut microbiota via amplicon-based 16S rRNA gene sequencing of the V3-V4 region in 54 patients with ET according to mutation status Janus-kinase 2 (*JAK2V617F*)-positive vs. *JAK2V617F*-negative patients with ET, and in 42 healthy controls (HC) were compared. The authors found that patients with ET have a distinct microbiota signature compared with HCs. There are some issues need to be addressed.

1. In this study, patients with ET were more often female. However, the gut microbiota is influenced by gender. How to rule out

the impact of gender on gut microbiota?

2. The group of JAK2V617-negative patients with ET was too small.
3. Which one or several are key microbes and the correlation between key microbes and clinical indicators.
4. Please check if the LDA score is faithful.
5. The overall analysis is too simple and insufficient.

Reviewer #3 (Comments for the Author):

There are some serious problems in the article: 1. The sample collection is not strict enough; 2, data analysis and discussion need to be further in-depth; Why not do an in-depth analysis at the species level? 3. Why not build an early risk assessment model based on gut flora?

Reviewer #4 (Comments for the Author):

The topic of the manuscript is relatively novel. However, there are also some problems: 1. Diet and age have a great impact on intestinal flora, which should be described in detail by the author. If these factors are too different, the accuracy of the data will be affected. 2. The exclusion criteria for samples are also too simple, and other conditions can affect the flora such as some disease conditions.

Staff Comments:

Preparing Revision Guidelines

Please return the manuscript within 60 days; if you cannot complete the modification within this time period, please contact me. If you do not wish to modify the manuscript and prefer to submit it to another journal, please notify me of your decision immediately so that the manuscript may be formally withdrawn from consideration by Microbiology Spectrum.

Response to Reviewers

Reviewer comments: Spectrum00662-23 (Pronounced Gut Microbiota Signatures in Patients with *JAK2V617F*-positive Essential Thrombocythemia)

Dear Editor and Reviewers,

We thank the reviewers for their reviews of the manuscript and helpful suggestions and comments.

Reviewer #2 (Comments for the Author):

In the present paper, the gut microbiota via amplicon-based 16S rRNA gene sequencing of the V3-V4 region in 54 patients with ET according to mutation status Janus-kinase 2 (*JAK2V617F*)-positive vs. *JAK2V617F*-negative patients with ET, and in 42 healthy controls (HC) were compared. The authors found that patients with ET have a distinct microbiota signature compared with HCs. There are some issues need to be addressed.

1. In this study, patients with ET were more often female. However, the gut microbiota is influenced by gender. How to rule out the impact of gender on gut microbiota?

- We are very grateful for this Reviewer's valuable comment. We agree, we cannot entirely rule out the effect of gender, and we are well aware of the problem. Since the number of patients included in this analysis is small (n= 11 male, n=25 female), we did not have enough power to investigate the sex-stratified differences. We do however, recognize the problem and have added the following as a limitation.

-Line 274 "Diet and sex-differences are known to influence the gut microbiota. Unfortunately, data on patients' diet was not available for this study. Furthermore, the number of patients were too small to investigate sex-differences. (44,45)."

2. The group of *JAK2V617*-negative patients with ET was too small.

- We are aware, that the group of *JAK2V617*-negative patients with ET is small, however ET is a rare cancer with an incidence between 0.38 to 1.7 per 100,000 (1). Moreover, a large proportion of patients with ET are positive for the *JAK2V617* mutation (2), limiting the number of *JAK2V617*-negative patients available for data analysis. This is also stated in the limitation section.
- Line 265 to 269 "Furthermore, the number of *JAK2V617F*-negative patients with ET was small (limiting the statistical power) and heterogeneous with other mutations (*CALR*, *MPL*) or triple-negative. We do not know whether the *CALR* mutation in itself affects the gut microbiota, and larger studies would be needed to elucidate a possible effect."

3. Which one or several are key microbes and the correlation between key microbes and clinical indicators.

- Initially we performed a correlation analysis, assessing the association between the abundance of different bacterial taxa and paraclinical parameters. But due to the differences in the gut microbiota between patients positive for the *JAK2V617F*-mutation and patients negative for the *JAK2V617F*-mutation, we evaluated the correlations in patients positive for the *JAK2V617F*-mutation only, and in healthy controls. However, very few significant correlations were found and the analysis thus excluded from the manuscript (please see Figure 1). If this analysis is required for publication, we will gladly add this to the manuscript.

Figure 1. Correlation analysis of paraclinical parameters in patients with ET positive for the *JAK2V617F*-mutation and in Healthy Controls.

Spearman's rank correlations on the continuous variables leukocyte count, thrombocyte count (trc), hemoglobin level (hgb), sedimentation reaction (SR), body mass index (BMI), lactate dehydrogenase (LDH), estimated glomerular filtration rate (eGFR), and Neutrophil-to-lymphocyte ratio (NLR) in ET patients with the *JAK2V617F*-mutation and healthy controls at genus level. Only correlations <-0.2 and > 0.2 are shown. Blue indicates a negative correlation and red a positive correlation. Asterisks indicate the following levels of significance: *, $p<0.05$; **, $p<0.01$; and ***, $p<0.001$.

4. Please check if the LDA score is faithful.

- Dear reviewer,

The LDA score in LEfSe analysis represents an effect size for each taxon's discriminatory power between two groups.

According to Segata et al., (3) the ranking of taxa according to $\log_{10}(\text{LDA score})$ is most relevant to assess the importance of a taxon, independent of absolute LDA scores.

Hence, we are confident that we can trust the results of our LEfSe analysis with regards to LDA scores.

5. The overall analysis is too simple and insufficient.

- We characterized the gut microbiota using amplicon sequencing of the 16S rRNA gene, and have used the most common analysis for this kind of data (alpha diversity, beta diversity). Moreover, we performed a differential abundance analysis using not only LEfSe, but we also confirmed our results by MaAsLin 2 analysis (4), which is a novel and advanced method tailored to microbiota data. Furthermore, we included a cluster analysis. The choice of excluding correlation analyses is explained above. We did consider doing PICRUSt2 analysis to explore metabolic pathways, however on 16S data it is only suggestive, and the sample size is small. Fragmented species level annotation is a known limitation and integral part of 16S rRNA sequencing. For further in-depth analyses at the species level, it would require omitting a large proportion of data not annotated to species level. In specific cases, it was possible to obtain a species annotation. e.g. *Faecalibacterium prausnitzii*, which we made use of.

-Line 129” The majority (89%) of reads affiliated with the genus *Faecalibacterium* had a species annotation as *F. prausnitzii*, the rest were unspecified species.”

To evade the limitation of fragmented species annotation, we performed selected analyses on amplicon sequence variant (ASV) level, which results from the dada2 pipeline and provides an even higher taxonomic resolution (single nucleotide) than species level.

Reviewer #3 (Comments for the Author):

There are some serious problems in the article:

1. The sample collection is not strict enough;

- Dear reviewer, thank you for your comments. The study was performed according to The STORMS checklist for human microbiome research (5). Please see

doi.org/10.6084/m9.figshare.22060499

All samples were collected and frozen within 6 hours and all samples were handled the exact same way. Furthermore, “Patients were excluded in case of any change of cytoreductive treatment within 3 months, use of antibiotics within 2 months, treatment with glucocorticoids, pregnancy, or inability to understand oral or written information.” as stated in line 297 to 299.

2. data analysis and discussion need to be further in-depth; Why not do an in-depth analysis at the species level?

- Dear reviewer. We characterized the gut microbiota using amplicon sequencing of the 16S rRNA gene, and have used the most common analysis for this kind of data (alpha diversity, beta diversity). Moreover, we performed a differential abundance analysis using not only LEfSe, but we also confirmed our results by MaAsLin 2 analysis (4), which is a novel and advanced method tailored to microbiota data. Furthermore, we included a cluster analysis. The choice of excluding correlation analyses is explained above. We did consider doing PICRUSt2 analysis to explore metabolic pathways, however on 16S data it is only suggestive, and the sample size is small. Fragmented species level annotation is a known limitation and integral part of 16S rRNA sequencing. For further in-depth analyses at the species level, it would require omitting a large proportion of data not annotated to species level. In specific cases, it was possible to obtain a species annotation. e.g. *Faecalibacterium prausnitzii*, which we made use of. Line 129 “The majority (89%) of reads affiliated with the genus *Faecalibacterium* had a species annotation as *F. prausnitzii*, the rest were unspecified species.”

To evade the limitation of fragmented species annotation, we performed selected analyses on amplicon sequence variant (ASV) level, which results from the dada2 pipeline and provides an even higher taxonomic resolution (single nucleotide) than species level.

3. Why not build an early risk assessment model based on gut flora?

- While that is an excellent idea, our sample size is too small (due to the limited number of patients available). But definitely something to pursue in the future with a larger patient cohort with serial assessments of gut microbiota and hard endpoints on patient survival.

Reviewer #4 (Comments for the Author):

The topic of the manuscript is relatively novel. However, there are also some problems:

1. Diet and age have a great impact on intestinal flora, which should be described in detail by the author. If these factors are too different, the accuracy of the data will be affected.

- Dear reviewer, thank you for your comments. While age definitely can impact the gut microbiota, no significant difference in age was seen between the groups. You are right, diet could potentially impact the gut microbiota. Unfortunately, data on patients' diet was not available for this study. It is now stated in the limitation section.
- Line 274 "Diet and sex-differences are known to influence the gut microbiota. Unfortunately, data on patients' diet was not available for this study. Furthermore, the number of patients were too small to investigate sex-differences.

2. The exclusion criteria for samples are also too simple, and other conditions can affect the flora such as some disease conditions.

- Thank you for your valid point, many factors are known to impact the gut microbiota such as age, diet, treatment with antibiotics, but also non antibiotic drugs, metformin, and many more, but also other factors, such as change of diet (6–9). Patients with ET in general are elderly, and are known to have a higher level of co-morbidities compared with HCs (10). Hence, if we made the inclusion criteria stricter, and only included patients with ET and no other comorbidities it would not represent the general ET patient population. Furthermore, very few patients would be eligible for inclusion. All comorbidities were assessed using Charlson Comorbidity Index (CCI). CCI provides a weighted score taking both the number and severity (1-6) of 19 different comorbidities into account (11). Our patients with ET had a median CCI at 1 (range 0–4) which is still quite low.

References:

1. Moulard O, Mehta J, Fryzek J, Olivares R, Iqbal U, Mesa RA. 2014. Epidemiology of myelofibrosis, essential thrombocythemia, and polycythemia vera in the European Union. *Eur J Haematol* 92:289–297.
2. Baxter EJ, Scott LM, Campbell PJ, East C, Fourouclas N, Swanton S, Vassiliou GS, Bench AJ, Boyd EM, Curtin N, Scott MA, Erber WN, Green AR. 2005. Acquired mutation of the tyrosine kinase JAK2 in human myeloproliferative disorders. *The Lancet* 365:1054–1061.
3. Segata N, Izard J, Waldron L, Gevers D, Miropolsky L, Garrett WS, Huttenhower C. 2011. Metagenomic biomarker discovery and explanation. *Genome Biol* 12:R60.
4. Mallick H, Rahnavard A, McIver LJ, Ma S, Zhang Y, Nguyen LH, Tickle TL, Weingart G, Ren B, Schwager EH, Chatterjee S, Thompson KN, Wilkinson JE, Subramanian A, Lu Y, Waldron L, Paulson JN, Franzosa EA, Bravo HC, Huttenhower C. 2021. Multivariable association discovery in population-scale meta-omics studies. *PLoS Comput Biol* 17:e1009442.
5. Mirzayi C, Renson A, Furlanello C, Sansone SA, Zohra F, Elsafoury S, Geistlinger L, Kasselmann LJ, Eckenrode K, van de Wijgert J, Loughman A, Marques FZ, MacIntyre DA, Arumugam M, Azhar R, Beghini F, Bergstrom K, Bhatt A, Bisanz JE, Braun J, Bravo HC, Buck GA, Bushman F, Casero D, Clarke G, Collado MC, Cotter PD, Cryan JF, Demmer RT, Devkota S, Elinav E, Escobar JS, Fettweis J, Finn RD, Fodor AA, Forslund S, Franke A, Furlanello C, Gilbert J, Grice E, Haibe-Kains B, Handley S, Herd P, Holmes S, Jacobs JP, Karstens L, Knight R, Knights D, Koren O, Kwon DS, Langille M, Lindsay B, McGovern D, McHardy AC, McWeeney S, Mueller NT, Nezi L, Olm M, Palm N, Pasolli E, Raes J, Redinbo MR, Rühlemann M, Balfour Sartor R, Schloss PD, Schriml L, Segal E, Shardell M, Sharpton T, Smirnova E, Sokol H, Sonnenburg JL, Srinivasan S, Thingholm LB, Turnbaugh PJ, Upadhyay V, Walls RL, Wilmes P, Yamada T, Zeller G, Zhang M, Zhao N, Zhao L, Bao W, Culhane A, Devanarayan V, Dopazo J, Fan X, Fischer M, Jones W, Kusko R, Mason CE, Mercer TR, Sansone SA, Scherer A, Shi L, Thakkar S, Tong W, Wolfinger R, Hunter C, Segata N, Huttenhower C, Dowd JB, Jones HE, Waldron L. 2021. Reporting guidelines for human microbiome research: the STORMS checklist. *Nat Med* 27:1885–1892.
6. Zhang Q, Hu N. 2020. Effects of Metformin on the Gut Microbiota in Obesity and Type 2 Diabetes Mellitus. *Diabetes Metab Syndr Obes* 13:5003–5014.
7. Hu X, Li H, Zhao X, Zhou R, Liu H, Sun Y, Fan Y, Shi Y, Qiao S, Liu S, Liu H, Zhang S. 2021. Multi-omics study reveals that statin therapy is associated with restoration of gut microbiota homeostasis and improvement in outcomes in patients with acute coronary syndrome. *Theranostics* 11:5778–5793.
8. Maier L, Pruteanu M, Kuhn M, Zeller G, Telzerow A, Anderson EE, Brochado AR, Fernandez KC, Dose H, Mori H, Patil KR, Bork P, Typas A. 2018. Extensive impact of non-antibiotic drugs on human gut bacteria. *Nature* 555:623–628.

9. Gomaa EZ. 2020. Human gut microbiota/microbiome in health and diseases: a review. *Antonie Van Leeuwenhoek* 113:2019–2040.
10. Frederiksen H, Szépligeti S, Bak M, Ghanima W, Hasselbalch HC, Christiansen CF. 2019. Vascular diseases in patients with chronic myeloproliferative neoplasms – Impact of comorbidity. *Clin Epidemiol* 11:955–967.
11. Charlson ME, Pompei P, Ales KL, MacKenzie CR. 1987. A new method of classifying prognostic comorbidity in longitudinal studies: Development and validation. *J Chronic Dis* 40:373–383.

July 17, 2023

Dr. Christina Schjellerup Eickhardt-Dalbøge
The Regional Department of Clinical Microbiology, University Hospital of Region Zealand
The Regional Department of Clinical Microbiology, University Hospital of Region Zealand
Ingemannsvej 46, stuen
Slagelse 4200
Denmark

Re: Spectrum00662-23R1 (Pronounced Gut Microbiota Signatures in Patients with *JAK2V617F*-positive Essential Thrombocythemia)

Dear Dr. Christina Schjellerup Eickhardt-Dalbøge:

Thank you for submitting your manuscript to Microbiology Spectrum. As you will see your paper is very close to acceptance.

Please comply with ASM data deposition policy - All sequence data needs to be publicly available.

As these revisions are quite minor, I expect that you should be able to turn in the revised paper in less than 30 days, if not sooner. If your manuscript was reviewed, you will find the reviewers' comments below.

When submitting the revised version of your paper, please provide (1) point-by-point responses to the issues raised by the reviewers as file type "Response to Reviewers," not in your cover letter, and (2) a PDF file that indicates the changes from the original submission (by highlighting or underlining the changes) as file type "Marked Up Manuscript - For Review Only". Please use this link to submit your revised manuscript. Detailed instructions on submitting your revised paper are below.

Link Not Available

Sincerely,

Jennifer Auchtung

Reviewer comments:

Reviewer #2 (Comments for the Author):

In this revised article, authors have made a proper response to my viewpoints.

Preparing Revision Guidelines

- Point-by-point responses to the issues raised by the reviewers in a file named "Response to Reviewers," NOT IN YOUR

COVER LETTER.

- Upload a compare copy of the manuscript (without figures) as a "Marked-Up Manuscript" file.
- Each figure must be uploaded as a separate file, and any multipanel figures must be assembled into one file.
- Manuscript: A .DOC version of the revised manuscript
- Figures: Editable, high-resolution, individual figure files are required at revision, TIFF or EPS files are preferred

Please return the manuscript within 60 days; if you cannot complete the modification within this time period, please contact me. If you do not wish to modify the manuscript and prefer to submit it to another journal, please notify me of your decision immediately so that the manuscript may be formally withdrawn from consideration by Microbiology Spectrum.

Dear Jennifer Auchtung Editor, Microbiology Spectrum.

Thank you for editing Spectrum00662-23R1 (Pronounced Gut Microbiota Signatures in Patients with *JAK2V617F*-positive Essential Thrombocythemia).

- Please comply with ASM data deposition policy - All sequence data needs to be publicly available.
 - The sequence data for patients with ET, anonymized, and are now available at the European Nucleotide Archive (ENA)(PRJEB58224).
 - Due to the European General Data Protection Regulations (GDPR), and present lack of approval from IRB to share data from participants, the dataset on HCs cannot for now be shared on ENA ". If investigators would like to access data, please contact Dr. Christina Ellervik or Dr. Christina Eickhardt-Dalbøge.

July 18, 2023

Dr. Christina Schjellerup Eickhardt-Dalbøge
The Regional Department of Clinical Microbiology, University Hospital of Region Zealand
The Regional Department of Clinical Microbiology, University Hospital of Region Zealand
Ingemannsvej 46, stuen
Slagelse 4200
Denmark

Re: Spectrum00662-23R2 (Pronounced Gut Microbiota Signatures in Patients with *JAK2V617F*-positive Essential Thrombocythemia)

Dear Dr. Christina Schjellerup Eickhardt-Dalbøge:

Your manuscript has been accepted, and I am forwarding it to the ASM Journals Department for publication. You will be notified when your proofs are ready to be viewed.

Sincerely,

Jennifer Auchtung
Editor, Microbiology Spectrum
